# Venomous Noodles: The Evolution of Toxins in Nemertea through Positive Selection and Gene Duplication

**DOI:** 10.3390/toxins15110650

**Published:** 2023-11-12

**Authors:** Gabriel Gonzalez Sonoda, Eric de Castro Tobaruela, Jon Norenburg, João Paulo Fabi, Sónia C. S. Andrade

**Affiliations:** 1Departamento de Genética e Biologia Evolutiva, IB-Universidade de São Paulo, São Paulo 05508-090, Brazil; sonoda198@gmail.com; 2Instituto Butantan, São Paulo 05503-900, Brazil; 3Faculdade de Ciências Farmacêuticas, Food Research Center (FoRC), Universidade de São Paulo, São Paulo 05508-080, Brazil; erictobaruela@usp.br (E.d.C.T.); jpfabi@usp.br (J.P.F.); 4Smithsonian Institution, Washington, DC 20024, USA; norenbur@si.edu

**Keywords:** ribbon worms, dN/dS, gene duplication, molecular evolution, Cytotoxin-A, We used a proteo-transcriptomic approach to detect toxins in two nemertean species. Additionally, our study made the first investigation of the molecular evolution of toxins, showing, for the first time, evidence for positive selection in toxin genes in this phylum. Lastly, we discuss how gene duplications may have impacted the evolution of toxin genes in Nemertea.

## Abstract

Some, probably most and perhaps all, members of the phylum Nemertea are poisonous, documented so far from marine and benthic specimens. Although the toxicity of these animals has been long known, systematic studies on the characterization of toxins, mechanisms of toxicity, and toxin evolution for this group are scarce. Here, we present the first investigation of the molecular evolution of toxins in Nemertea. Using a proteo-transcriptomic approach, we described toxins in the body and poisonous mucus of the pilidiophoran *Lineus sanguineus* and the hoplonemertean *Nemertopsis pamelaroeae*. Using these new and publicly available transcriptomes, we investigated the molecular evolution of six selected toxin gene families. In addition, we also characterized in silico the toxin genes found in the interstitial hoplonemertean, *Ototyphlonemertes erneba*, a meiofaunal taxa. We successfully identified over 200 toxin transcripts in each of these species. Evidence of positive selection and gene duplication was observed in all investigated toxin genes. We hypothesized that the increased rates of gene duplications observed for Pilidiophora could be involved with the expansion of toxin genes. Studies concerning the natural history of Nemertea are still needed to understand the evolution of their toxins. Nevertheless, our results show evolutionary mechanisms similar to other venomous groups.

## 1. Introduction

Ribbon worm is the common name for animals belonging to the phylum Nemertea, a group with more than 1300 valid species [1]. The synapomorphy of the group is presence of an eversible proboscis housed in a cavity called the rhynchocoel. With few exceptions, nemerteans are free-living predators or scavengers inhabiting the marine benthos [2]. Recently, it was proposed that the phylum should be divided into three classes: Hoplonemertea, Palaeonemertea, and Pilidiophora, the latter comprising the order Heteronemertea and the family Hubrechtellidae [3]. The reciprocal monophyly of Pilidiophora, Hoplonemertea, and Palaeonemertea was recovered in a phylogenomics approach by Andrade et al. [4]. Although their soft body makes them look vulnerable and inoffensive, animals from this phylum possess toxins that may be used for predation and defense. The first known record of a ribbon worm was in 1555 [5], and includes the first description of its venomous nature. However, more detailed and systematic studies of the toxin molecules were not published until the 1930s [6,7]. Since then, more studies have addressed the biochemical activity of these toxins [8,9,10,11,12,13] and their role in the natural history of Nemertea [14,15].

Toxins are defined as substances produced by living organisms that cause deleterious effects in other organisms (named targets) exposed to them [16]. Due to their potential to act as pharmacological agents and pesticides, the isolation and description of these substances are common practices. The ecological role of the toxins in Nemertea has been largely deduced from field and laboratory observations coupled with biochemical assays [11,15,17,18]. It is known that the proboscis plays a fundamental role in aiding the delivery of these toxins. Hoplonemerteans bear stylets on the proboscis that pierce a target’s body wall, creating a wound through which the toxin-containing substance enters the target organism [14,19]. In the case of the monostiliferous Hoplonemertea, toxin may be actively introduced via the pumping action of a specialized, muscular, mid-proboscis bulb [20]. Most Palaeonemertea and Pilidiophora lack any rigid piercing structure. These worms rely on the physical contact of the proboscis coated in a poisonous mucus to envenom their prey [19]. In addition, toxins can be found in the mucus surrounding the animal’s body, acting as a potential defense mechanism against predators. Such mucus was demonstrated to contain peptides and other substances with neurotoxic and cytotoxic properties [12,13,21].

Both proteinaceous [9,12,21] and non-proteinaceous [8,22,23,24,25] toxic compounds have been isolated from the mucus of different nemertean species. There is evidence that some of these toxins have potential application as pesticide [13].

Non-proteinaceous toxins identified in Nemertea include pyridine alkaloids and Tetrodotoxin (TTX). The first nemertean pyridine alkaloids were firstly described in the Hoplonemertea *Paranemertes peregrina* [6]. Later, more pyridine alkaloids were characterized from other hoplonemernteans and characterized as neurotoxic to invertebrates [8,23] and, in a smaller degree, mammals [22]. Recently, it was proposed that these toxins evolved before the the monostiliferan–polystiliferan divergence [24]. TTX, a strong paralytic compound that blocks sodium channels [26], was detected in all three Nemertea orders [25], but was found to be specially abundant in the Palaeonemertea *Cephalothrix simula* [25]. Studying the distribution of TTX in this species, it was proposed that this toxin is obtained via food uptake [27,28,29]. Moreover, there is evidence that bacteria associated with *Cephalothrix simula* produce TTX [30]. In a different species, potential adaptations in the Nemertea sodium channels were found, possibly explaining how these worms are resistant to Tetrodotoxin [31]. While non-protein toxins require analytical and biochemical characterization, proteo-transcriptomic approaches allow for the identification and evolutionary analyses of proteinaceous toxins, which will be the focus of this study. So far, the only nemertean proteinaceous toxins that have been successfully isolated and had their toxicity assayed were obtained from Pilidiophora specimens. However, Whelan et al. (2014) [32] and Vlasenko et al. (2022) [33] found putative toxins in the transcriptomes of the three classes of Nemertea. Recently, proteomic and transcriptomic approaches showed new potential proteinaceous toxins in the mucus and proboscis of hoplonemerteans [34,35]. These findings point toward various unexplored proteinaceous toxins in Nemertea, with their bioactivity potential yet to be investigated.

Despite this tantalizing potential of Nemertea as a hotbed of toxin evolution, the phylum has received little attention in reviews of metazoan toxins [32], while few groups, such as scorpions, snakes, spiders, and cone-snails, have been in the spotlight for studies concerning biology and the evolution of venoms [36]. These studies point to clear patterns in the evolution of toxin genes and their origins. The classical and main hypothesis for their origin is through the duplication of toxin genes, which may go under neofunctionalization [37,38]. The multilocus toxin gene families support this hypothesis in cone-snails [39,40,41,42], spiders [43], snakes [44,45,46,47], and scorpions [48,49,50]. Further, endogenous genes within toxin gene families evidence the role of the neofunctionalization of endogenous gene copies to the origin of toxins [44,46,47,50]. Also, other mechanisms have been proposed to explain the origin of these genes, including alternative splicing [51], modification of a physiological gene [52], and lateral gene transfer [43]. The multilocus toxin gene families also indicate that further gene duplication events are common in the evolutionary history of toxins. This might result from the diversification of the chemical arsenal of organisms through neofunctionalization, thereby increasing the number of potential molecular targets for a given venom, which may increase their fitness.

Another critical pattern identified in the evolution of toxins is the adaptive (or positive) selection in toxin gene families, inferred through the non-synonymous/synonymous mutation ratio (dN/dS), as observed in the toxin genes of cone-snails [39,42], scorpions [49,53], snakes [45,54], and spiders [43]. As described by Van Valen’s Red Queen hypothesis [55], a coevolution process between producer and target might trigger this particular pattern [38]. Positive selection in these gene families could also be a by-product of gene duplications followed by neofunctionalization, as the new gene copy product may evolve into a protein with an adaptive role [56].

The main goal of this study was to investigate the molecular processes behind toxin evolution in Nemertea and to search for patterns described in the evolution of toxins in other metazoans. By investigating toxin evolution in this poorly studied phylum and expanding the number of taxa investigated so far, our results will contribute to a better and less biased understanding of the origins and forces that drive the evolution of toxin genes in metazoans. To that end, we have sequenced transcriptomes of three Nemertea species, the cosmopolitan pilidiophoran *Lineus sanguineus* (Rathke 1799), the hoplonemerteans *Ototyphlonemertes erneba* Corrêa 1950, and *Nemertopsis pamelaroeae* Mendes et al., 2021. In addition, proteomic approaches were used to validate the new putative proteinaceous toxin found. The use of multi-omic approaches and the availability of previously sequenced data from the three Nemertea classes made it possible to analyze this molecular evolution in Nemertea from an integrative perspective with an emphasis on Pilidiophora, shedding light on the evolutionary scenario of the toxins of an understudied and ecologically crucial group.

## 2. Results and Discussion

In this study, we investigated the molecular evolution of the toxins in the phylum Nemertea. Through a proteo-transcriptomic approach, using both published data and new transcriptomes, we were able to identify known and new putative toxins in the inspected taxa, as well as their homologous in Nemertea. Using a robust new phylogeny, we detected a higher gene duplication rate observed in Pilidiophora when compared to the remaining Nemertea classes that could be associated with the origin of evolutionary novelties in this group, such as new toxin genes. Upon closer inspection, we observed that orthogroups containing putative toxins presented more duplications in Pilidiophora than in Hoplonemertea and Palaeonemertea. In addition, evidence for positive selection was detected in six toxin genes, indicating an accelerated evolution comparable to the toxin genes of more studied venomous taxa. These results are the first to empirically support that the evolution of toxins in Nemertea is similar to the evolution of toxins in most Metazoans.

### 2.1. Transcriptome Assembly

After quality trimming, the raw Illumina reads were used to assemble the transcriptomes with TRINITY [57]. The RIN and the main assembly metrics from *Lineus sanguineus*, *Nemertopsis pamelaroeae,* and *Ototyphlonemertes erneba* are presented in Appendix A. The BUSCO [58] analysis showed that the assembled transcriptomes presented homologs for a great majority of the conserved metazoan gene dataset, indicating that the transcriptomes are highly diverse in terms of represented genes (Appendix A). The completeness of the downloaded transcriptomes ranged from 12% in *Tubulanus punctatus* to 97% in *Lineus viridis*. The low completeness from the *Lineus lacteus* is due to the fact that such data correspond to partial exome sequencing [59].

### 2.2. Phylogenomic Analysis

The assembled transcriptomes were used to infer a reliable species tree which was needed to better comprehend the history of gene duplications in toxin gene families. For this, the proteomes predicted from the assembled transcriptomes were clustered with OrthoFinder [60]. The number of represented orthogroups ranged from 1815 in *Tubulanus punctatus* to 104,753 in the *Lineus sanguineus* collected in Brazil. After occupancy filtering (Appendix A), the 5208 orthogroups were submitted to monophyly masking. We evaluated the compositional heterogeneity effect on our 310,946 residuals-aligned matrix. Based on the values distribution on a 95% confidence interval, the 271 orthogroups with the highest Relative Composition Frequency Variability (RCFV) were removed. The final matrix contained 299,880 aligned amino acids and 31 taxa.

The relationship of the major clades (Palaeonemertea, Pilidiophora, and Hoplonemertea) remained as found in previous phylogenetic inferences based on genetic data (Figure 1) [4,61]. Unlike Kvist et al. [62], we did not recover a paraphyletic Palaeonemertea (Figure 1). The topology of both the Hoplonemertea and Heteronemertea groups are very similar to the ones found in the literature, including the position of the collected taxa in the tree (i.e., *Lineus sanguineus*, *Nemertopsis pamelaroeae*, and *Ototyphlonemertes erneba*) [4,62]. The obtained phylogenomic tree suggests that *Riseriellus occultus* should be moved to the genus *Lineus*, which can also be observed in other works [4,62]. All these findings are supported by nodes with a 100% support value from the ultrafast bootstrap method.

### 2.3. Identified Putative Toxins

The identification of putative toxins was based on the sequence similarity to known toxins and their presence in the mucus proteome. Through the search by similarity, we found 227, 232, and 208 putative toxin transcripts for *Lineus sanguineus, Nemertopsis pamelaroeae,* and *Ototyphlonemertes erneba*, respectively (Appendix A). All these transcripts had a BLAST best hit in 74, 79, and 74 different proteins from the custom database, respectively, for *L. sanguineus*, *N. pamelaroeae,* and *O. erneba* (Appendix A). The best hits against the ToxProt database included toxins from animals of different phyla, such as Cnidaria, Mollusca (genus *Conus*), vertebrates (snakes and fishes), arthropods (spiders and chilopods), and Echinodermata (crown-of-thorns starfish). Whelan et al. [32] found fewer toxin transcripts in the transcriptomes of related species, probably due to differences in the toxin designation methodology. Our results are more similar to those presented by Vlasenko et al. [33], who also used the ToxProt database. The main differences consist in the transcripts aligning to the nemertean toxins added to our database.

Moreover, we identified an additional 12 transcripts for *L. sanguineus* as putative toxins for being present in the mucus proteomes and presenting features that indicate their role as toxins (Appendix A). These features include toxin domains such as cysteine knots (Pfam domain Toxin_35), PhTx neurotoxin family [63], ion-channel inhibitory toxin (Pfam_domain Toxin_12), and Na/K-Atpase Interacting protein [64] (Appendix A). These proteins were found in the mucus and contain domains similar to neurotoxins. Although these putative toxins were not characterized, peptides with neurotoxic proprieties were found in the defensive mucus of *Cerebratulus lacteus* [9] and *Lineus longissimus* [13], indicating a potential neurotoxic role for the putative toxins described in this study. The lack of significative alignments of these transcripts to our custom toxin database suggests that the *Lineus sanguineus* new putative toxins are not homologous to the toxins recently described for Hoplonemertea [34,35]. Moreover, it emphasizes how our understanding of the diversity of toxins in Nemertea is limited, supporting the importance of proteo-transcriptomic approaches to identifying new toxins.

The number of putative toxins found was similar between the three studied species. For the Hoplonemertea, most putative toxins were annotated to the custom Nemertea toxin database (Appendix A); meanwhile, for the *Lineus sanguineus*, a greater proportion corresponded to toxins from the ToxProt (Appendix A). This difference reflects the representation of the classes in each database: while most characterized toxins found in Heteronemertea have been curated and added to the ToxProt, the toxins identified by proteo-transcriptomic approaches in Hoplonemertea [34,35] are still absent from most public databases. Even so, as in other recent studies, our results suggest a considerable diversity of proteinaceous toxins in Hoplonemertea [34,35].

We can highlight the putative toxin transcripts identified as *Snaclec*, *Actitoxin,* and the *Plancitoxin-1* (similar to *U-nemertotoxin-1*) found in the Lineidae and both hoplonemerteans putative toxin transcripts (Appendix A); these were also found to be transcribed in the proboscis of *Amphiporus lactifloreus*. Both *Actitoxin* and *Plancitoxin-1* were also present in the defensive mucus proteome, suggesting an actual role in predation and defense in that species [34]. The finding of transcripts similar to these toxins in *Ototyphlonemertes erneba* (Appendix A) is, to our knowledge, the first report of toxins from meiofaunal species. Although their toxicity is yet to be confirmed, their expression in the proboscis of a related species indicates that these also act as toxins in *O. erneba*. Furthermore, transcripts similar to *Vulnericin* were identified for the first time in Nemertea in this meiofaunal species (Appendix A).

Correlations between the ecology and the venom of *Ototyphlonemertes erneba* can not be clearly established. We know from rare gut content observations that the species of *Ototyphlonemertes* may feed on annelids and crustaceans [65] (and JLN, pers. Obs.) and from Corrêa’s (1949) [66] method for capturing *Ototyphlonemertes* with freshly dead fish that they can be necrophagous. There is, however, no description of the role of venom in prey capture and defense for these meiofaunal animals. Nevertheless, the unusually wide diversity of proboscis structure [67] among the species of the genus suggests the possibility of strong prey preferences (JLN, pers. obs.). One might expect similar diversity of unique toxins among the species.

### 2.4. Proteomic Experiments

To confirm the translation of putative toxins identified in the transcriptome and to identify more putative toxins, body and mucus proteomic analyses were carried out using high-performance liquid chromatography with tandem mass spectrometry (HPLC-MS/MS). From the final set of 74 putative toxins from *Lineus sanguineus*, 17 were found in the mucus or body proteome, and 11 of the 79 putative toxins from *N. pamelaoreae* were found in the mucus and body proteomes of *Nemertopsis berthalutzae.* Toxins identified in the transcriptome, but absent from the protein samples, may have been obscured from the analyses by abundant structural and endogenous proteins present in the body of both animals. Nevertheless, considering the small size of the animals, the low yield of protein obtained in the mucus extraction, and potential post-translational modifications, these results are solid and validate the presence of potential toxins in the mucus, providing valuable information on the composition of this defensive secretion.

A total of 83,236 MS1 scans were obtained from all analyzed fractions of the four *Lineus sanguineus* bodies separately. The number of MS1 scans per fraction ranged from 13,837 to 13,951. From these scans, PEAKS Studio Xpro identified 2577 proteins in the body, 37 of which were annotated as putative toxins with areas higher than zero (Appendix A). For the *Nemertopsis berthalutzae* body, a total of 41,777 MS1 scans were obtained from fractions. The number of MS1 scans from fractions varied between 13,859 and 13,975. PEAKS Studio Xpro identified 576 proteins from the transcriptome of *Nemertopsis pamelaroeae* through these scans, 19 of which were considered toxins with areas higher than zero (Appendix A).

The only toxin peptide with an area higher than zero in the seawater control sample was an *Antistasin-like* peptide in *Lineus sanguineus* seawater, although Peaks did not find the same peptide in the mucus sample (Appendix A), which may indicate a potential misidentification exclusive to the seawater sample. For this reason, we did not consider any of the putative toxins as contaminants from the seawater. The mucus proteome from *Lineus sanguineus* and *Nemertopsis berthalutzae* yielded, respectively, 6945 and 6912 MS1 scans, while the control sample containing only seawater yielded 6900 scans. PEAKS Studio Xpro assigned these scans to 25 and 82 different proteins derived from the studied species transcriptome, respectively, for *L. sanguineus* and *N. berthalutzae*. Of these, 16 were identified as toxins in *L. sanguineus* (Appendix A) and 1 was identified as a toxin in *N. berthalutzae* (Appendix A).

The mucus proteome of the *L. sanguineus* and *N. berthalutzae* studies showed no sign of *Cytotoxin-A* or *U-nemertotoxin-1* homologs, contrasting with previous reports of toxins found in the mucus of *Parborlasia corrugatus* [12], *Cerebratulus lacteus* [10], and *Amphiporus lactifloreus* [34], respectively. Since transcripts homologous to *Cytotoxin-A* and matching peptides were respectively found in the transcriptome and body proteome of *Lineus sanguineus* (Appendix A), these toxins might have lost their defensive role and remain expressed only in the proboscis in this species, playing an important role in predation. Our results are the first to suggest different roles for these toxins in different species of Nemertea. Such differences could have evolved in response to diet changes and defense requirements (i.e., predators). Shifts in the composition of venom coupled with changes in its ecological role and delivery system are also present in spitting cobras, in which an increase in the cytotoxic activity of the venom is related to the defensive role of the spitted venom in causing pain [68]. Furthermore, snails from the genus *Conus* present distinct sets of toxins for defensive and predatory roles expressed in different parts of the venom duct [69]. It can be compared to the different sets of toxins being expressed in the body tegument and glandular epithelium of proboscis in Nemertea.

Other putative toxins found in the mucus of *L. sanguineus* were similar to *Antistasin*, a protein found in the salivary gland of leeches. *Antistasin* was found to be an anticoagulant and inhibitor of serine-proteases [70]. Using MEME v.5.5.1 [71], no anticoagulant motifs (obtained from [72]) were found on the *Antistasin* sequences. In non-blood-feeding invertebrates, antistasin and proteins with similar domains are thought to have a role in immune responses, though their role remains unclear [72]. Verdes et al. [35] found *antistasin* in the proboscis of the hoplonemertean *Antarctonemertes valida*, in which it likely is a predation toxin. Toxins homologous to serine protease inhibitors can also be found in the venom of sea anemones and cone-snails, some of which present a potassium-channel-blocking activity [73], though these are usually more similar to *kunitz* and not to *antistasins*.

Still, considering their capability of inhibiting serine-proteases, which often plays a digestive role in marine organisms [74], together with the lack of anticoagulant motifs in these sequences, the presence of antistasin in the defensive mucus of *Lineus sanguineus* might inhibit the digestion in eventual predators, rather than interfere with its hemostasis. In accordance with that, it was observed that the anemone *Metridium senile* kept in aquariums regurgitates ingested *Cerebratulus lacteus* (J. Norenburg, personal observations).

### 2.5. Selected Toxin Gene Families

Based on previous reports of toxicity and their presence in proteome, the following putative toxin genes were selected for the analyses of selection test and gene duplications: *Cytotoxin-A*, *scoloptoxin SD976-like*, *alpha-nemertide*, *beta-nemertide*, *U2-Agatoxin-like* (which we named *U-Nemertotoxin-3*), and a putative toxin showing low similarity to *alpha-Ktx* (blast bitscore = 28.5), *beta-defensin* (phmmer score = 23.2), and *myticin* (Pfam score = 16). Transcripts from *scoloptoxin SD976-like* were found in both *L. sanguineus* and *N. pamelaroeae* transcriptomes, while transcripts from the remaining toxins were exclusive to *L. sanguineus*. Peptides derived from the *scoloptoxin SD976-like* were found in the body proteome of both *L. sanguineus* and *N. berthalutzae*. Peptides from *Cytotoxin-A* were only found in the body proteome of *L. sanguineus* and the remaining were only found in the mucus proteome of *L. sanguineus* (Appendix A).

Final codon alignments for the *Cytotoxin-A*, *scoloptoxin-SD976-like*, *alpha-nemertide*, beta-nemertide, *U2-Agatoxin-like,* and *alpha-KTx-like* had, respectively, 598, 2058, 342, 273, 351, and 204, base pairs. These alignments and derived gene trees were used to perform selection tests and infer gene duplications by an algorithm of tree reconciliation.

#### 2.5.1. Cytotoxin-A

*Cytotoxin-A* are cytotoxic proteins that, so far, have only been isolated from Pilidiophora. *Cytotoxin-A* proteins were first isolated from *Cerebratulus lacteus* mucus [10]. A homologous toxin was isolated from *Parborlasia corrugatus* mucus and called *Parborlysin* [12], both of which were demonstrated to have cytolytic properties. Similar to Jacobsson et al. [13] and Vlasenko et al. [33], we found a putative transcript for *Cytotoxin-A* in the *Hubrecthella ijimaji* transcriptome, suggesting that such a toxin was present in the ancestor of Pilidiophora. However, these were excluded from further analysis for being too divergent from the remaining species toxins.

#### 2.5.2. Scoloptoxin SD976-like Protein

We identified scoloptoxins, namely the *Scoloptoxin SSD552* (uncharacterized) and *Scoloptoxin SSD976* (a Voltage-gated calcium channel inhibitor), both of which are cysteine-rich secretory proteins (CRISP) isolated from the chilopoda *Scolopendra subspinipes dehaani* [75], in the body transcriptomes and proteomes of *Lineus sanguineus* and *Nemertopsis pamelaroeae* (Appendix A). Also, some of the transcripts in the same orthogroup presented similarity to *latisemin*, a CRISP found in the venom of the sea snake *Laticauda semifasciata* (Appendix A). However, more experiments are required to determine if the expressed protein is a toxin.

An hmmscan analysis showed that many of these proteins contain different numbers of ShK-domain-like. This domain is similar to the *potassium channel blocker peptide ShK* described in the sea anemone *Stichodactyla helianthus* [76]. Although this domain has also been described in non-toxins [77,78], it often plays an important role in controlling potassium channels, as in the *MMP23* [78]. Moreover, a previous study on nemertean toxins reported a putative toxin containing the ShK domain [34], therefore the reported protein likely corresponds to a toxin.

#### 2.5.3. Alpha-Nemertide

The *alpha-nemertides* are peptides that target Na channels with selectivity for arthropods over mammal Na-channels, indicating a biotechnological potential as a pesticide [13]. The orthology inference showed that these peptides are mostly found in the genus *Lineus* (and *Riseriellus*), although a transcript was found in *Cerebratulus marginatus*. A thoroughly functional characterization of these toxins’ family described the effects of different *alpha-nemertides* in Na-channels. However, the role of these peptides in envenomation has yet to be fully elucidated [79].

#### 2.5.4. Beta-Nemertide

The *Beta-nemertides* were described along with the *Alpha-nemertide*; they share some homology with *Neurotoxin-BII*, indicating they may also act as a neurotoxin, but were not biochemically characterized [13]. Its presence and in the mucus of *Lineus longissimus* [13] and *Lineus sanguineus* further validate its role as a toxin.

#### 2.5.5. The U-Nemertotoxin-3: Agelenin-like Proteins, a New Toxin?

The *Lineus sanguineus* mucus presented peptides of a protein containing a inhibitor cystine knot domain (ICK), a domain present in other known toxins [80,81,82] including the *Alpha-nemertide* [13]. This same protein presents considerable similarity to *Agelenin* (Appendix A). *Agelenins* are peptides isolated from the venom of the spider *Agelena opulenta* [83], which was demonstrated to produce instantaneous paralysis in crickets [84]. The presence of an ICK domain and similarity to other known neurotoxins suggests it should be a new neurotoxin, which will be named *U-Nemertotoxin-3,* following the nomenclature of Nemertea toxins proposed by von Reumont et al. [34] and the system proposed by King et al. [85]. It is noteworthy that the nomenclature of nemertean toxins has not been standardized yet. Transcripts from the *U-Nemertotoxin-3* gene family were found only in the *Lineus* and *Riseriellus* genera.

#### 2.5.6. Alpha-KTx-Like/Beta-Defensin-Like/Myticin-like

Lastly, we found an intriguing putative toxin in the mucus proteome, which had the scorpion potassium channel toxin *alpha-KTx 26.2* (e-value = 1.7) as the best blast hit and *Beta-defensin* as the phammer best hit (e-value = 0.0023). Also, the hmmscan indicates that it resembles *Myticin*, an antimicrobial peptide found in mussels [86]. Nonetheless, the hmmscan also indicates that two transcripts in this gene orthogroup showed low similarity to a domain that confers K+ Channel blocking activity [87] contained in the family of the aforementioned scorpion toxin. This protein shows similarities to both a K+ Channel blocker and a pore-forming protein such as the *Beta-defensin* [88]. In fact, the employment of the defensin domain is recurrent in venoms from other animals, including neurotoxins from scorpions [89]. Although we considered this a putative toxin, a non-excluding hypothesis is that it acts as an antimicrobial peptide, forming pores in pathogenic bacteria that may infect the exposed tegument of Nemertea.

Besides being found throughout the Pilidiophora, homologs were also found in transcriptomes from both *Cephalothrix* species examined, but were absent from the remaining Palaeonemertea and Hoplonemertea.

We selected the gene families most likely to be toxins for further analyses. The toxicity of *cytotoxin-A* and the *alpha-nemertides* has been well described in previous studies [10,12,13,79]. Although no biochemical description of the *beta-nemertides* has been made to our knowledge, its presence in the mucus with similarity to the *Neurotoxin-BII,* which has been biochemically characterized [9], made Jacobsson et al. [13] consider it a toxin. Although toxicity assays are necessary to assert the role of the remaining gene families as toxins, we can still hypothesize their role as toxins based on their similarity to other toxins and their presence in the mucus, a defensive secretion.

### 2.6. Selection Tests in Toxins

To better understand how the sequence of these toxins evolved, we applied selection tests using models accounting for codons with heterogeneous evolution rates. In the alignments of all six analyzed genes, the presence of sites under positive selection was statistically significant under the site model for both the HyPhy [90] (Figure 2, Figure 3, Figure 4, Figure 5, Figure 6 and Figure 7) and CODEML [91] (Appendix A) analyses (*p* < 0.05). Additionally, in each gene family, at least one codon was detected to be under positive selection using two or more methods (Table 1). Moreover, the branch site model, which, additionally, allows for heterogeneity between the branches of the gene tree, was applied for the *Cytotoxin-A*, *scoloptoxin SD976-like,* and *alpha-nemertide* gene families, which presented gene duplications in nodes containing three or more species. *Cytotoxin-A* and *alpha-nemertide* had one branch presenting sites under positive selection (Table 1).

As the main driver of selection in toxin genes is the toxin target (the prey or predator that will be affected by such a toxin), the substitution in these positively selected sites could lead to changes in the toxin activity and specificity as an evolutionary response to the different selective pressures imposed by the ecological challenges faced by each species. In other words, these amino acid substitutions could be an evolutionary response to the changes in the composition of its targets. Many of these sites under selection fell within toxin domains, supporting this hypothesis (Appendix A).

For all tested toxins, except for the *Scoloptoxin SD976-like*, we found evidence of positive selection within clades delimited by duplications (branch site model) or without duplications (M7/M8, FUBAR, and FEL). This indicates that natural selection led to toxin divergence between different species. Put in another way, changes in these sites in toxin genes may be being driven by a selective pressure for the toxin to work better against the targets of each species. Positive selection between species has been previously described in toxins [92,93].

At the same time, we observed sites under positive selection for the *Cytotoxin-A*, *Scoloptoxin SD976-like, alpha-nemertides*, and *beta-nemertides* gene families presenting duplications. The existence of sites found to be under positive selection by the site models (M7/M8, FEL. FUBAR), but not by the branch site model, could be explained by natural selection favoring the divergence of paralogs toxins. Gene duplications with positive selection between the copies may lead to an increase in venom complexity, which may allow for an increase in the diversity of the potential molecular targets of these toxins via neofunctionalization. Diversifying potential molecular targets may help overcome eventual resistances evolved in the target. Alternatively, it may allow for the inclusion of new species in the diet, or to act as a defense mechanism against new potential predators. Such positive selection after duplications has been previously described in toxins [42,53,94] and physiological mammal genes [56]. The detection of pervasive selection is particularly useful for identifying sites that may lead to subtle changes in protein activity while maintaining overall function. However, while in *Cytotoxin-A* no sites under positive selection were detected using FUBAR and FEL, an increased number of sites under episodic positive selection was identified with MEME (Table 1). This implies that different sites might experience positive selection in different lineages. Grueber et al. [95] attributed episodic positive selection in innate immunity avian genes to parasite–host co-evolution. The same logic might be applied to a nemertean–target co-evolution, as different lineages might be challenged by different targets, requiring a fast adaptation for a new ecological role. The prevalence of episodic over pervasive positive selection was also observed in cnidarians toxins [96]. The lack of data on the 3D structure of *Cytotoxin-A* hinders our understanding on how these substitutions may affect toxin activity.

To test how toxin evolution is related to the natural history of Nemertea is a challenging task. As our data show, different species might employ the same toxin in different contexts (i.e., defense and predation). Also, the natural history of ribbon worms needs to be better elucidated due to their secretive way of life; thus, the knowledge of actual prey and predators for different species of Nemertea is scarce.

Either way, these hypotheses leading to target diversification and specialization may guide future searches for toxins that could have pharmacological applications. The expected small changes in the toxin caused by these sites under positive selection may be of interest to the development of pharmaceuticals and pesticides, as they are likely a result of selective pressure for changes in toxin activity and selectivity.

### 2.7. Gene Duplications in Nemertea

To determine the number of toxin gene copies and their genomic locations, we aligned *Lineus longissimus* toxin transcripts to the same species genome. For *Cytotoxin-A*, five loci were found in tandem (Appendix A), whereas *scoloptoxin SD976-like* transcripts were mapped to six different genomic locations. When mapping only the transcripts retained with MaxAlign, this number decreases to two (Appendix A). For the *alpha-nemertide*, the number of mapped regions was four, three of which were in the same genome scaffold. The remaining three putative toxin transcripts mapped to two genomic regions each, although some alignments presented no intron, indicating potential pseudogenization events (Appendix A). These alignments not only revealed the number of copies for these genes in the genome, but also showed that many of these copies were in tandem. Evidence for tandem duplication of toxin genes is also observed in spiders [97] and snakes [98,99,100,101].

The tree reconciliation method also provided insights into the number and phylogenomic depth of duplications in toxins in Nemertea (Figure 2, Figure 3, Figure 4, Figure 5, Figure 6 and Figure 7; Appendix A). By combining this method with the genome alignment, we observed that the expansion of the analyzed toxin genes is not a feature restricted to the *Lineus longissimus*, but is spread throughout the Pilidiophora lineage. The presence of transcripts derived from these older duplications in the transcriptome of different species is evidence that these gene copies are present and expressed in many taxa.

The genomic alignments and the reconciled tree results were also shown to be complementary, allowing for a deeper understanding of the evolution of toxin gene families. For instance, the four *Cytotoxin-A* transcripts we assembled mapped to five different genomic regions (Appendix A). However, these transcripts belong to only one of the three clades descending from gene duplications in the *Cytotoxin-A* phylogeny (Figure 2), while transcripts from the remaining clades were not found in *L. longissimus*. This could result from a lack of expression from copies belonging to the remaining clades or differences in the number of gene copies between species. For example, in *Notospermus geniculatus*, 11 *Cytotoxin-A* loci were found [102], more than twice the number we reported in *L. longissimus*.

Duplications have an essential role in the evolution of toxins, either by giving birth to new genes that can be modified into toxins or by increasing the number of genetic copies of those toxins [38]. The finding of more putative toxin genes in *Lineus sanguineus* proteome, in addition to the fact that proteinaceous toxins have only been isolated from Pilidiophora [5], raises the question of whether this class has experienced more duplications than the remaining Nemertea. Using CAFE and OrthoFinder, we estimated the number of gene duplication events in each node in the species tree to answer this question. The number of duplications in each node inferred using CAFE ranged from 7 to 3311 (Figure 8), while in OrthoFinder, this number ranged from 32 to 5334 (Appendix A).

CAFE also allowed us to estimate the rate of expansion and contraction events per gene family per million years (ƛ) in different branches of the tree. Restricting the whole tree to a single value, we found a ƛ of 7.23 × 10^−4^ (Table 2). In the two ƛ scenario, Pilidiophora had a higher ƛ (1.44 × 10^−3^) than the remaining taxa (7.23 × 10^−4^), similar to the one in the single ƛ scenario. Adding a third ƛ for the three *Lineus sanguineus* clade did not cause major changes in the other two ƛ. Adding an error model to deal with assembly errors did not change the ƛ drastically for Pilidiophora or the remaining taxa, but slightly changed the ƛ for *Lineus sanguineus*. All of the scenarios allowing more than one lambda were significantly more likely than the global lambda scenario (*p* < 0.01) (Table 2).

Our results (Appendix A) show that nodes within Pilidiophora have more duplications than the remaining Nemertea. Gene duplications result in redundant proteins, which may experience neofunctionalization and lead to evolutionary novelties [103,104,105]. Alternatively, new copies allow for simultaneous transcription, potentially increasing the expression of the duplicated gene [106]. Either way, such an increased rate of gene duplications in Pilidiophora could have resulted in genetic novelties, such as the origin of new toxins or the expansion of toxin gene families. In fact, we found an elevated number of duplications in toxins containing orthogroups within Pilidiophora (Appendix A).

Besides Pilidiophora, the palaeonemertean genus *Cephalothrix* also displays an increased number of gene duplications (Appendix A), specially in orthogroups containing toxins (Appendix A), when compared to the remaining Nemertea. This, coupled with the findings of Whelan et al. [32], who described a high diversity of toxins in the genus, may indicate that this group is a good candidate for future proteinaceous toxin prospection. Interestingly, the species *Cephalothrix simula* was found to have high concentrations of Tetrodotoxin in its body [25].

Gene duplications leading to venom diversity as an evolutionary response to the number of possible targets that could be related to more generalist diets that includes a diversity of polychaetes, oligochaetes, and other nemerteans in Pilidiophora [19]. In contrast, Hoplonemerteans tend to present a specialized diet [19], such as the animals from the genus *Nemertopsis,* which have only been documented to feed on barnacles [107]. Specialized diets would not impose selective pressure toward diversifying predatory toxins since the current specialized arsenal efficiently subdues the prey. However, this does not explain the lack of defensive toxins in the mucus of *N. berthalutzae*, since it inhabits the same kind of habitat as *Lineus sanguineus* and could encounter the same predators. A possible explanation is that, in some groups, non-proteinaceous toxins play a major role in envenomation, whereas peptides are prevalent in others. This could be the reason that the characterized proteinaceous toxins from Nemertea were all isolated from Pilidiophora, while many non-proteinaceous toxins have been isolated and characterized in Hoplonemertea [5].

Multigene families are known for the toxin genes of different venomous animals [39,40,41,42,43,44,45,46,47,48,49], and here we show that this seems to be the case for Nemertea as well. As discussed earlier, new copies may be advantageous at first as they may allow for the simultaneous transcription of more than one copy, resulting in augmented gene expression [106]. Moreover, it may also facilitate the diversification of the venom via the neofunctionalization of one of the copies, which could result in positive selection [108]. The diversity of targets may determine the fitness change resulting from the diversification of the venom. In snakes, for instance, there seems to be a positive correlation between the complexity of venom and the phylogenetic distance of its preys [109,110,111].

## 3. Conclusions

In this work, we prospect for putative toxins in three Nemertea species using a proteo-transcriptomic approach. We report a high diversity of putative toxin genes in *Lineus sanguineus*, *Nemertopsis pamelaroeae,* and *Ototyphlonemertes erneba*, with orthologs in other species. To the best of our knowledge, this is the first study to systematically assess the molecular evolution of toxin genes in Nemertea, accounting for gene duplications and selection tests. We show that the toxin genes observed for these animals evolve similarly to those described in snakes, spiders, scorpions, and cone-snails. As in these groups, Nemertea toxin genes may have experienced positive selection and gene duplications. Nevertheless, there is still much to be elucidated on the biochemical characteristics of these toxins and the role of these toxins in the natural history of ribbon worms. Summing up, our study and other recent publications on the topic reveal an extensive diversity of toxins produced by these animals, which up until now have been relatively neglected. Beyond any doubt, there is a lot to learn from these curious animals, and what we know so far is but a small piece of this intriguing puzzle.

## 4. Materials and Methods

### 4.1. Sampling

Specimens of *Lineus sanguineus*, *Nemertopsis pamelaroeae,* and *N. berthalutzae* Mendes et al., 2021 were collected in beds of the oyster *Crassostrea* sp. along the southeast Brazilian coast (Table 3). Large portions of the beds were collected using a hammer and a putty knife. The fragments were placed in trays containing seawater at room temperature, left for two hours until the individuals crawled to the water surface, and then removed with a thin brush. *Ototyphlonemertes erneba* specimens were collected by dripping water in a tray with collected sediment, following Corrêa’s protocol [66]. The species were morphologically identified using Envall and Norenburg’s [67] key characteristics list. All samples were preserved in RNAlater (Table 3). Animals were sampled under permits issued by Institute Chico Mendes (ICMBio), protocol numbers 55,701 and 67,004.

### 4.2. RNA Extraction and Sequencing

Total RNA was extracted using Tri-Reagent (Ambion), using the RNA extraction and purification protocols described in Riesgo et al. [112].

For *Nemertopsis pamelaroeae* and *Lineus sanguineus* extractions, a whole individual was used for RNA extraction. For *Ototyphlonemertes erneba* transcriptomes, extraction was performed using a pool of 12 individuals, since a single sample did not yield enough RNA. The RNA quality and quantity was assessed in a Qubit fluorometer v.3 (Thermo Fisher Scientific, Waltham, MA, USA), and samples with ratios between 1.8 and 2.2 were considered pure. Sample integrity was confirmed with BioAnalyser equipment (Agilent Technologies Inc., Santa Clara, CA, USA). A cDNA library was produced from each sample using the TruSeq RNA Library Preparation V2 kit (Illumina Inc., San Diego, CA, USA). Libraries were clustered and sequenced on the Illumina HiSeq 2500 platform with 2 × 100 bp paired-end, using a TruSeq SBS V3 kit (Illumina Inc., Thermo Fisher Scientific), at the Laboratório de Biotecnologia Animal, Escola Superior de Agricultura “Luiz de Queiroz”—Universidade de São Paulo (ESALQ-USP).

Raw reads from 21 Nemertea taxa comprising 5 Palaeonemertea, 10 Pilidiophora, and 6 Hoplonemertea species transcriptomes were downloaded from the NCBI database using the SRA toolkit [113] and are presented in Appendix A. These were selected based on the availability of paired-end Illumina reads from the body transcriptome. Two additional *Lineus sanguineus* transcriptomes were included. Transcriptomes from all individuals were used for phylogeny inference, gene duplication analysis, and selection tests (Appendix A). Assembly completeness was evaluated using the database metazoa_odb9 from BUSCO [58].

### 4.3. Transcriptome Assembly and ORF Prediction

Prior to assembling, the overall quality of the reads was evaluated using the software FastQC v0.11.3 [114]. Reads with average quality under 24 phred score, adapters and putative contaminant reads were filtered out using the software Seqyclean v.1.10.09 [107] and the UniVec (https://www.ncbi.nlm.nih.gov/tools/vecscreen/univec/, acessed on 12 October 2023) as the putative contaminant database. The transcriptomes were assembled using the software Trinity v2.8.4 [115]. TransDecoder v5.5.0 [116] was used to predict the translated proteins and their respective coding sequences (CDSs), with a cut-off of a minimum of 30 amino acids for the prospection of toxin genes, selection tests, and duplication analyses of toxin genes; the cut-off was 100 amino acids for the phylogenomic analysis and overall gene duplication analysis (Appendix A). The 30-amino-acid cut-off was defined considering the previous report of toxins smaller than 100 amino acids, such as the alpha and beta-nemertides [13]. The higher cut-off for the phylogenomics and overall gene duplication analysis was chosen to avoid orthogroups with short sequences, potential mis-assemblies, and false open reading frames (ORFs).

### 4.4. Phylogenomic Analysis

To proceed with the phylogenomic and molecular evolutionary analysis, the predicted proteins for the transcriptomes of each species were assigned to orthogroups using the software OrthoFinder v2.3.3 [60] with default parameters.

The orthogroups were filtered according to their occupancy, and only groups with at least 26 taxa (~80% occupancy) were kept. Each orthogroup was aligned using MAFFT L-INS-i v.7.407 [117], and each gene tree was obtained using IQTree v.1.6.12 [118]. Node support was evaluated with 100 ultrafast bootstrap replicates. FASTA-formatted files were trimmed with TrimAl v1.2 to account for alignment uncertainty, with a gap threshold of 80% and conserving a minimum of 20% of the original alignment [119]. Monophyly masking was performed with an iterative paralogy pruning procedure using PhyloTreePruner [120].

The package BaCoCa v.1.105r [121] was used to estimate the Relative Composition Frequency Variability (RCFV), which measures the absolute deviation from the mean for each amino acid and for each taxon and sums these up over all amino acids and all taxa [122]. Based on the value distributions, partitions with a high degree of compositional heterogeneity were filtered out.

The concatenated matrix was analyzed with IQ-Tree, using LG4X+G as evolution model and 1000 bootstrap replicates. Representatives of the phyla Annelida and Mollusca were chosen as outgroups based on previous studies on metazoa relationships [123]. The species chosen as outgroups were the Annelida *Phyllochaetopterus* sp., *Myzostoma seymourcollegiorum*, and *Capitella teleta*, and the Mollusca *Solemya velum*, *Lottia gigantea,* and *Monodonta labio* (Appendix A).

### 4.5. Toxin Identification

The transcriptomes of the three sequenced species (pilidiophoran *Lineus sanguineus*, hoplonemerteans *Nemertopsis pamelaroeae,* and *Ototyphlonemertes erneba*) were assessed for transcripts encoding toxins. Their predicted proteomes (see *Transcriptome assembly and ORF prediction section*; Appendix A) were annotated to a database containing SwissProt [124], ToxProt (last update on 18 April 2023) [125], and a custom database using BLAST V2.9.0 [126]. The custom database was created with toxin sequences identified in previous Nemertea proteo-transcriptomic studies [13,34,35]. Sequences with the best hits against sequences from the custom Nemertea database or the ToxProt and with e-values lower than 10 × 10^−5^ were submitted to predict signal peptides with Phobius V 1.01 [127]. We observed that imprecise predictions of the start codon caused Phobius to assign signal peptides as a trans-membrane domain near the N-terminal of the protein. To deal with that, we looked for the closest methionine to the trans-membrane domain in these proteins and set it as the new start codon, trimming the remaining N-terminal amino acids. For these sequences, we re-analyzed them using Phobius. The resulting toxin-matching proteins containing evidence for signal peptides were considered putative toxins. Additionally, as we expect a high abundance of toxins in the mucus proteome, we annotated the proteins detected in the mucus samples (see below) with more permissive parameters (e-value < 10). We manually reviewed the annotation of the proteins found in the mucus to identify additional putative toxins.

Identified putative toxins were further annotated against the same database using phammer (HMMER suite v3.3 [128]). Finally, their domains were predicted using hmmscan, as well as from HMMER suite and the Pfam 33.1 database [129].

As this study focused on the toxins from Nemertea, we filtered out prokaryotic toxins. Also, although *Perivitelline-2* is present in ToxProt, hits to this toxin were filtered out as it is mostly known for its role in the nutrition and defense of eggs ([130], although see [131]).

### 4.6. Proteomic of Mucus and Whole Individuals

The presence of the predicted proteins in either the body or the mucus proteome validated the transcriptomic results. Also, we considered the presence of the predicted putative toxins in the mucus as evidence for its role as a toxin. We performed proteomic experiments with four *Lineus sanguineus* and two *Nemertopsis berthalutzae*, also collected in southeastern Brazil (Table 3). *N. berthalutzae* was used as no new fresh samples from *N. pamelaroeae* were found in the fieldwork. As sister species to *N. pamelaroeae*, both belonging to the *Nemertopsis bivittata* complex [132], we expected similar defensive toxins between these species, which coexist on the coast and occupy the same habitat [132], meaning they might face the same predators and feed on the same prey. Before mucus extraction, a sample of the water in which animals from each species rested for 10 min was collected as a control. After that, animals were stimulated to secrete mucus via repeated aspiration and expiration from a plastic pipette and manual shaking of a 2 mL vial containing approximately 1.5 mL of seawater. Three cycles of ten aspirations followed by 10 s of manual shaking were performed for each animal, with a 10 min interval between each cycle. Animals were also carefully scraped with a hypodermic needle to remove the additional mucus from the skin. Afterward, the samples, the seawater containing the mucus, and the control were separately stored in liquid nitrogen, then lyophilized and stored at −20 °C (protocol adapted from H. Andersson, personal communication).

Lyophilized mucus samples were resuspended in milli-Q water, and excess salt was removed via size exclusion chromatography using the gravity protocol of columns packed with Sephadex^®^ G-25 Medium (PD-10, Cytiva, Marlborough, MA, USA), as in Jacobsson et al. [13]. The desalted samples were lyophilized again before dithiothreitol (DTT) reduction, iodoacetamide (IAA) alkylation, and digestion with Trypsin and Lys-C (Trypsin/Lys-C Mix, Mass Spec Grade Promega) for 14 h at 37 °C.

For the body proteome, individuals with dry mass ranging between 1.1 mg and 3.5 mg were manually homogenized with micropistilles in a solution of LAEMMLI with 10% beta-mercaptoethanol. The final concentration was 7.2 μg of animal/μL. In order to avoid the more abundant structural proteins overshadowing the toxins, samples were fractionated in an 12.5% SDS-PAGE following [133]. The resulting gel was cut in three fragments accordingly to protein content visualized using Coomassie blue G staining and were submitted to in-gel digestion. The used protocol was a modified version of Shevchenko et al. (2006) [134], in which proteins were reduced (DTT) and alkylated (IAA) before the digestion with Trypsin and Lys-C (Trypsin/Lys-C Mix, Mass Spec Grade Promega) for 14 h at 37 °C. Lastly, the digested peptides of both mucus and bodies were desalted using C18 ZipTips columns (Millipore) and concentrated using a CentriVap (Labconco Corporation, Kansas City, MI, USA).

Peptides were redissolved in 0.1% formic acid (FA), injected in a C18 pre-column (Acclaim PepMap RSLC Nano-Trap column; 3 μm, 100 Å, 75 μm × 20 mm, Thermo Fisher Scientific) and separated on a C18 analytical column (Acclaim PepMap RSLC column; 2 μm, 100 Å, 75 μm × 150 mm, Thermo Fisher Scientific) using a linear gradient from 5% buffer A (80% acetonitrile, 0.1% FA) to 95% for 120 min, with buffer B containing 0.1% FA. The flow rate was set to 350 nL/min and the oven temperature to 50 °C. The QToF Impact II Bruker mass spectrometer (Bruker Daltonics, Bremen, Germany) was interfaced to the nanoLC system (Thermo Scientific UltiMate™ 3000 RSLCnano system) with the CaptiveSpray nanoBooster source (Bruker Daltonics, Billerica, MA, USA) using acetonitrile as dopant. Liquid chromatography coupled to mass spectrometry data were acquired using a Data Dependent Acquisition (DDA) scheme. Briefly, an MS scan at scan speed of 200 ms was followed by 20 MS/MS fragment scans of the most intense precursors (50 ms) in a total cycle time of 1.2 s. The mass range of the MS scan was set from *m/z* 150 to 2200. The isolation of precursor ions was performed using an *m/z* isolation window of 2.0 and a dynamic exclusion of 0.4 min. The collision energy was adjusted between 23 and 65 eV as a function of the m/z value. All samples were analyzed in random order.

PEAKS Studio Xpro (V10.6 PEAKS Team) was used to query scans against proteomes containing proteins with 30 amino acids or more (Appendix A), predicted from the transcriptomes of *Lineus sanguineus* or *Nemertopsis pamelaroeae*, in addition to the SwissProt dataset of proteins to search for contamination. Precursor mass search was set to monoisotopic, parent mass error tolerance was set to 25 ppm, fragment mass error tolerance was set 0.1 Da, and maximum FDR was set to 5%. The enzyme was set to Trypsin/Lys-C, and the maximum missed cleavage sites were set to three. For the mucus analyses, we considered peptides as contaminants if their peptides area in the seawater sample was more significant than in the mucus sample.

### 4.7. Trees and Alignments for Toxin Orthogroups

We only considered putative toxins validated in the proteomes for the evolutionary analysis. From these, we selected six putative toxins of *Lineus sanguineus* based on their presence in the transcriptomes of other species and on previous reports of these toxins.

Orthogroups containing these toxins and their homologous genes were selected. The translated amino acid sequences of these orthogroups were replaced by their respective CDS, previously inferred with TransDecoder. Redundant CDs within each orthogroup were filtered out using cd-hit-est v4.8.1 [135], with 100% similarity. The remaining sequences were aligned using codon with Muscle at MEGA-X [136]. We manually removed highly divergent and incomplete sequences for toxins found in the mucus. As the orthogroups containing the toxins found in the body had more sequences, MaxAlign v1.1 [137] was used to automatically remove sequences while keeping maximum gap-free alignment area. Maximum likelihood trees for these alignments were determined using RAxML v8.2.12 [138] rapid bootstrap analysis (N = 100, model = GTRGAMMA). The resulting trees were midpoint-rooted.

### 4.8. Selection Test

The orthogroups alignments obtained in the section “Trees and alignments for toxin orthogroups” and the gene trees reconciliation obtained in the section “Gene duplication in toxins” were tested for positive selection using ETE 3 [139], which employs CODEML, from the package PAML4 [91] and HyPhy V 2.5.51 [90]. To test for positive selection in the whole orthogroup, we tested the alignment for positive selection using site models. To detect pervasive positive selection in sites across the genes’ phylogeny, we compared CODEML M7 and M8 LTR and employed HyPhy FEL [140] and FUBAR [141]. Episodic positive selection in sites was detected with MEME [142]. Also, subsets of three or more sequences identified as originating from a duplication event were marked for a branch site selection test with CODEML models bsA1 and bsA. By marking these subsets of sequences, we tested for positive selection in each subset defined by duplication events. For CODEML results, statistical significance of the models was asserted by comparing the log likelihood ratio (LTR) of the null and alternative models of the site and branch-site models to a chi-squared distribution of one and two degrees of freedom, respectively. Also, for CODEML, Bayes Empirical Bayes was used to infer posterior probability of a site being positively selected (*p* > 0.95).

### 4.9. Gene Duplications in Toxin Orthogroups

To detect gene duplications, all gene trees were compared to the phylogenomic species tree. We assumed that orthogroups determined using OrthoFinder contained predicted proteins that shared homology, including orthologs and paralogs [60]. It implies that each sampled transcriptome containing more than one transcript in the orthogroup has experienced gene duplication. To detect gene duplications, gene tree reconciliation analyses were performed using DLCpar v2.0.1 [143], which uses dynamic programming to find the most parsimonious set of gene duplications, losses, and deep coalescence events that explains the observed data. As this analysis is based on transcriptomic data, it is not possible to distinguish gene duplications in one species from isoforms derived from alternative splicing of the same gene. So, duplication events leading to two sequences of the same species were counted out. Also, transcriptomic data does not allow for distinguishing between gene losses and the absence of transcription during sample fixation. Therefore, the gene loss results were not accounted for.

Since transcriptomic data can lead to a false interpretation of the number of genetic copies due to post-transcriptional variations, such as splicing, we validated these results using a combination of genomic and transcriptomic data. For this, *Lineus longissimus* transcripts (Appendix A) contained in toxin orthogroups were aligned to the genome of *L. longissimus* (PRJEB45185; [144]) using blat v. 36 [145] with default parameters. Alignments with introns bigger than 20 kb were filtered out. Alignments were visualized with IGV-Integrative Genomics Viewer v2.14.1 [146] and the R package GVIZ [147]. We defined the gene copy number as the different genomic regions where the transcripts aligned.

### 4.10. Nemertea Gene Duplication Events

Overall estimates of gene duplication were performed using OrthoFinder, which infers gene duplications by comparing a species tree given by the user (in this case, the phylogenomic tree) with rooted gene trees, inferred using the OrthoFinder ‘msa’ algorithm with default options. Such comparison is achieved through a hybrid algorithm of DLCpar and species overlap [60]. Estimates of the contribution of toxin gene families to these duplication events were assessed with the duplication counting on orthogroups containing the obtained putative toxins at each node of the species tree.

To further understand the expansion and contraction history of genes in the phylogeny of Nemertea, an estimation of the birth–death rate of a given gene family per millions of years (ƛ), using CAFE, v4.2.1 [148] was performed. To avoid false duplications inferred by misassembled short contigs or poorly predicted ORFs, we used the predicted proteomes with proteins containing 100 or more amino acids (Appendix A). Also, to avoid the misinterpretation of families with multiple isoforms, only the longest, and hence more informative, isoform proteins (identified using Trinity) were included in the CAFE analysis. These passed through an all vs. all blast and were clustered using MCL ([149], inflation value = 2.0). Clusters with 100 or more gene copies were filtered out to avoid gene families with high variance of copy numbers, as suggested by the author [148].

CAFE was used to estimate one scenario with one global ƛ value for the whole tree, another scenario allowing two different ƛ (one for Pilidiophora and another for the remaining taxa), and a third scenario with three ƛ (one for the *Lineus sanguineus* clade, one for the remaining Pilidiophora, and one for the remaining taxa). This third scenario aimed to reduce any bias resulting from the inclusion of three individuals of the same species. The third scenario was also recalculated with an error model accounting for assembling errors. This model accounting for errors was finally used to estimate the expansion and contraction events in each node of the phylogenomic tree. To test if allowing more than one ƛ in the tree increased the model’s likelihood, twice the difference between the log likelihood of the global ƛ scenario and the multi ƛ scenario (2 × (lnLikelihoodglobal − lnLikelihoodmulti)) was compared to a null distribution of likelihood ratios (2 × (lnLikelihoodglobal − lnLikelihoodnull)) obtained via 194 null simulations (excluding -inf values) of the data set with a single ƛ for the whole tree, from which *p*-values were estimated [148].

## Figures and Tables

**Figure 1 toxins-15-00650-f001:**
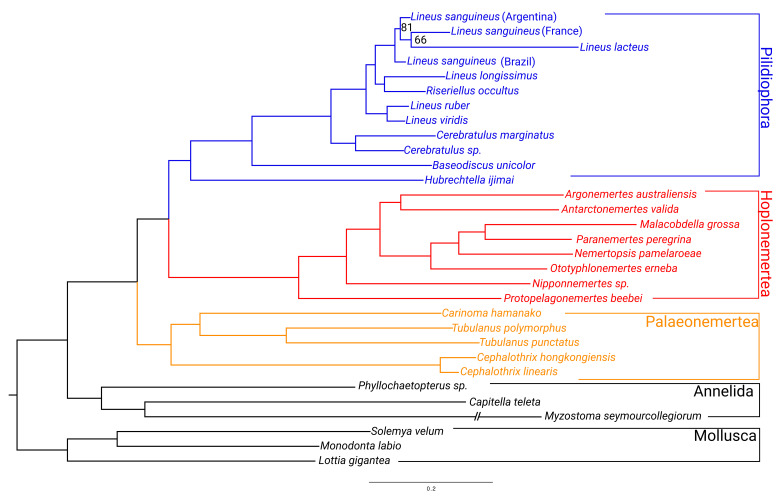
Phylogenomic tree of the Nemertea phylum obtained from 5208 genes (see Section 4). The tree was rooted to Mollusca. All the nodes have 100% bootstrap support except for those indicated.

**Figure 2 toxins-15-00650-f002:**
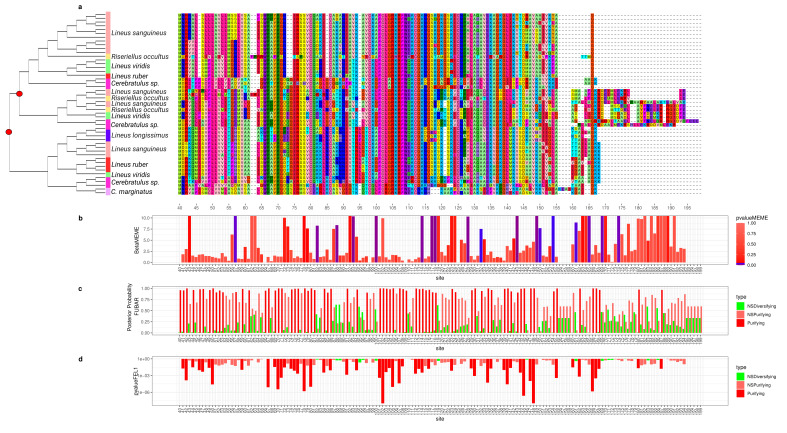
Evolutionary analyses for *Cytotoxin-A* orthogroup. (**a**) On the left, the reconciled phylogenetic tree is presented, with nodes attributed to duplications events indicated by red circles. On the right is the aligned protein sequences from the Cytotoxin-A orthogroup. For clearness, only part of the alignment is represented. For each position in the alignment, the probability of episodic positive selection was calculated with MEME (**b**); the probability of being under pervasive positive selection was calculated with a Bayesian approach with FUBAR (**c**) and with a probabilistic approach with FEL (**d**).

**Figure 3 toxins-15-00650-f003:**
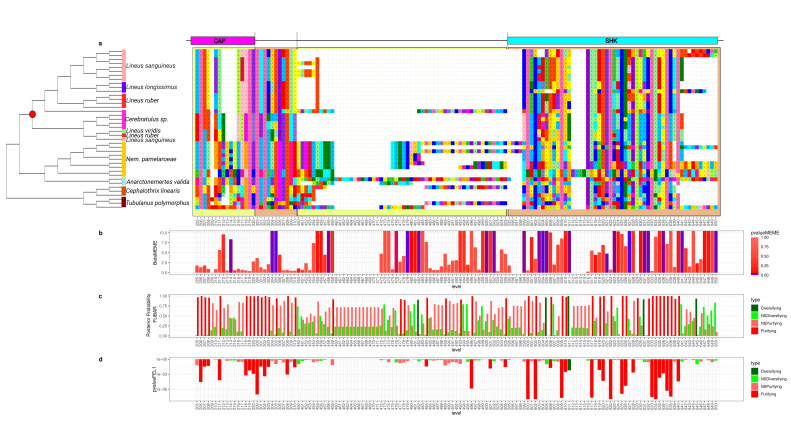
Evolutionary analyses for *Scoloptoxin-SD976* orthogroup. (**a**) CAP: cysteine-rich secretory protein domain. Shk: ShK-domain-like. The brown and yellow squares in the background are used to represent different parts of the protein. For clearness, only part of the alignment is represented. For each position in the alignment, the probability of episodic positive selection was calculated with MEME (**b**); the probability of being under pervasive positive selection was calculated with a Bayesian approach with FUBAR (**c**) and with a probabilistic approach with FEL (**d**).

**Figure 4 toxins-15-00650-f004:**
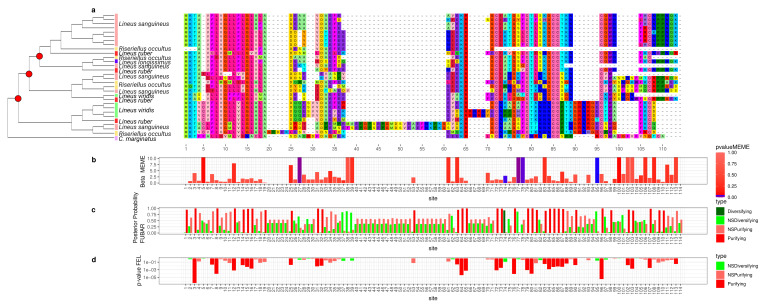
Evolutionary analyses for *Alpha-nemertide* orthogroup. (**a**) For clearness, only part of the alignment is represented. For each position in the alignment, the probability of episodic positive selection was calculated with MEME (**b**); the probability of being under pervasive positive selection was calculated with a Bayesian approach with FUBAR (**c**) and with a probabilistic approach with FEL (**d**).

**Figure 5 toxins-15-00650-f005:**
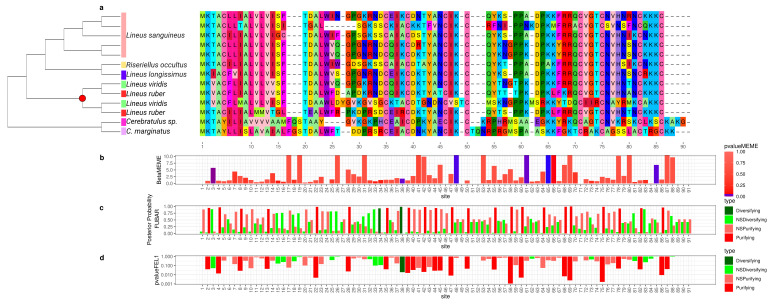
Evolutionary analyses for *Beta-nemertide* orthogroup. (**a**) For clearness, only part of the alignment is represented. For each position in the alignment, the probability of episodic positive selection was calculated with MEME (**b**); the probability of being under pervasive positive selection was calculated with a Bayesian approach with FUBAR (**c**) and with a probabilistic approach with FEL (**d**).

**Figure 6 toxins-15-00650-f006:**
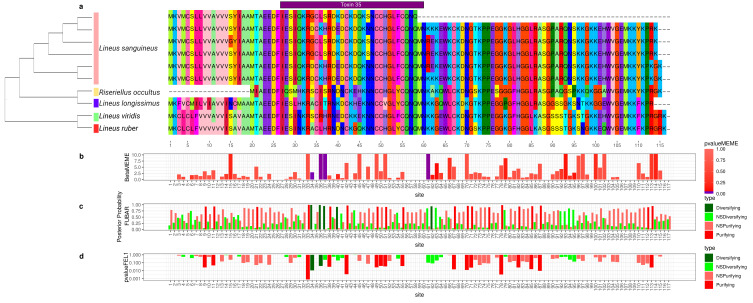
Evolutionary analyses for *U-Nemertotoxin-3* orthogroup. (**a**) Toxin 35: Toxin_35 Pfam domain (toxin with inhibitor cystine knot ICK or Knottins). For clearness, only part of the alignment is represented. For each position in the alignment, the probability of episodic positive selection was calculated with MEME (**b**); the probability of being under pervasive positive selection was calculated with a Bayesian approach with FUBAR (**c**) and with a probabilistic approach with FEL (**d**).

**Figure 7 toxins-15-00650-f007:**
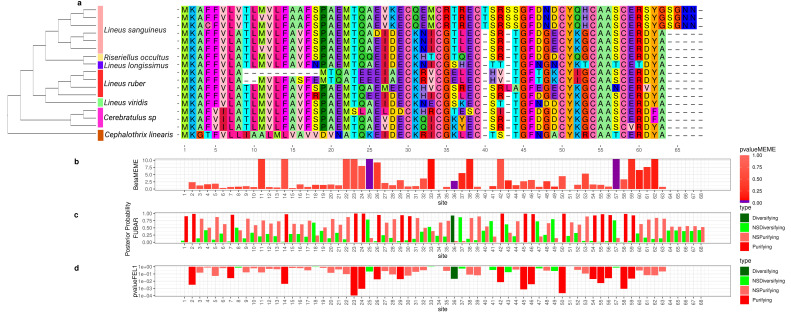
Evolutionary analyses for alpha-Ktx/Beta-defensin/myticin orthogroup. (**a**) For clearness, only part of the alignment is represented. For each position in the alignment, the probability of episodic positive selection was calculated with MEME (**b**); the probability of being under pervasive positive selection was calculated with a Bayesian approach with FUBAR (**c**) and with a probabilistic approach with FEL (**d**).

**Figure 8 toxins-15-00650-f008:**
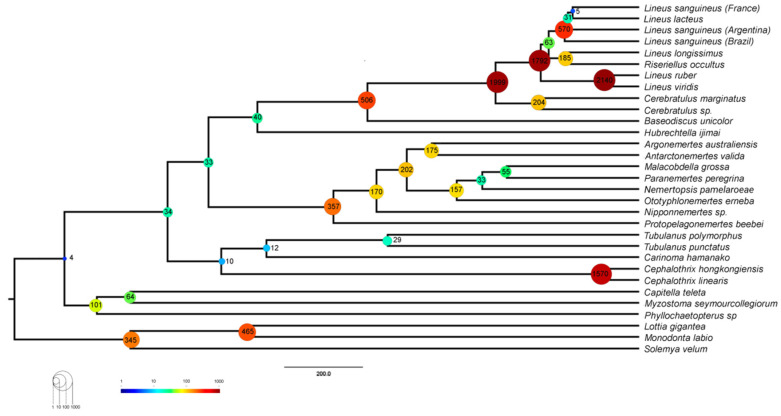
Number of expansion events in each node of the species tree obtained with CAFE analysis allowing for the three different lambdas and including an assembly error model. Numbers are plotted to the ultrametric tree used in the CAFE analysis.

**Table 1 toxins-15-00650-t001:** Results from selection tests. Sites detected via CODEML had BEB posterior probability superior to 0.95. “Common” line indicates sites identified as under positive selection using at least two methodologies. NA: Not applicable, as these genes were not tested under the branch site model. NS: Not significative.

Toxin	Positive Selection Detection Method	Sites under Selection
Cytotoxin	M8/M7 (BEB)	146
	BsA/BsA1 (BEB)	154
	MEME	57; 82; 88; 93; 100; 114; 117; 118; 128; 132; 143; 149; 150; 154; 161; 165; 169; 174
	FEL	NS
	FUBAR	NS
	Common (at least 2)	154
Scoloptoxin	M8/M7	481; 482; 604; 605; 611; 624; 631; 647; 649
	BsA/BsA1	NS
	MEME	26; 33; 36; 37; 38; 40; 58; 74; 123; 214; 277; 304; 458; 479; 482; 500; 502; 599; 603; 604; 605; 611; 621; 623; 627; 631; 632; 636; 641; 650
	FEL	611;
	FUBAR	482; 605; 611; 631; 645
	Common (at least 2)	482; 604; 605; 611; 631
Alpha-Nemertide	M8/M7	77
	BsA/BsA1	63; 77
	MEME	27; 74; 77; 78; 95
	FEL	NS
	FUBAR	74
	Common (at least 2)	77,74
Beta-Nemertide	M8/M7	26; 31
	BsA/BsA1	NA
	MEME	3; 38; 48; 61; 65; 85
	FEL	38
	FUBAR	34; 38
	Common (at least 2)	38
U-Nemertotoxin-3	M8/M7	36; 37
	BsA/BsA1	NA
	MEME	34; 36; 37; 61
	FEL	34; 36
	FUBAR	34; 36; 37; 40; 62
	Common (at least 2)	34; 36; 37
Myt-alpha-ktx-like	M8/M7	37
	BsA/BsA1	NA
	MEME	25; 36; 57
	FEL	36
	FUBAR	36
	Common (at least 2)	36

**Table 2 toxins-15-00650-t002:** CAFE results. Lambda (estimate of gene duplication and contractions per gene family per unit of time) for each scenario. The formula (2×(lnLglobal−lnLmulti)) is used to test two different models by using their log likelihood. This value was compared to the values obtained in 194 null simulations. The number of null simulations with lnL lower than the formula was the considered *p*-value. lognL: logn of likelihood; lnLglobal: logn of likelihood of single lambda model; lnLmulti: logn of likelihood of multi lambda models. NA: non-available.

Model	Global Lambda (or Remaining Species)	Pilidiophora Lambda	*Lineus* Lambda	lognL	(2 × (lnLglobal − lnLmulti))	Lowest (2 × (lnLglobal − lnLsimulation)) (N= 194)
1 ƛ (Global)	7.23 × 10^−4^	NA	NA	−1.03 × 10^6^	NA	−7.65
2 ƛ Heteronemertea Remaining	7.23 × 10^−4^	1.44 × 10^−3^	NA	−9.73 × 10^5^	−1.19 × 10^5^	
3 ƛ *Lineus sanguineus* Heteronemertea Remaining	7.23 × 10^−4^	1.44 × 10^−3^	4.11 × 10^−3^	−9.44 × 10^5^	−1.76 × 10^5^	
1 ƛ Global + Error model	7.23 × 10^−4^	NA	NA	−1.03 × 10^6^	NA	
3 ƛ *Lineus sanguineus* Heteronemertea Remaining+ error model	5.51 × 10^−4^	1.34 × 10^−3^	8.55 × 10^−4^	−1.02 × 10^6^	−2.54 × 10^4^	

**Table 3 toxins-15-00650-t003:** Species collected and their respective collection local and substrate. *: Used for RNA seq. +: Used for mucus proteomics. &: Used for body proteomics.

Species	Locality	Latitude	Longitude	Substrate	Number of Individuals
*L. sanguineus* *	Praia do Cabelo Gordo—SP	−23.828088	−45.4232	*Crassostrea* sp. *bank*	1
*L. sanguineus* +&	Praia do Araçá—SP	−23.812840	−45.408216	*Crassostrea* sp. *bank*	2
*L. sanguineus* &	Peró—RJ	−22.822844	−41.969951	*Crassostrea* sp. *bank*	2
*N. berthalutzae* +&	Praia Grande—SP	−23.824663	−45.412764	*Crassostrea* sp. *bank*	2
*N. pamelaroeae* *	Praia do Jabaquara—RJ	−23.211346	−44.714191	*Crassostrea* sp. *bank*	1
*O. erneba* *	Praia da Vila—SP	−23.777234	−45.358997	Sand sediment	12

## Data Availability

The data used in this study are deposited at SRA under BioProjects PRJNA952238, PRJNA952238, and PRJNA952238.

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
