# Peer review of "Venomous Noodles: The Evolution of Toxins in Nemertea through Positive Selection and Gene Duplication"

_toxins, 2023, doi:10.3390/toxins15110650_

Round 1

Reviewer 1 Report

Thanks for the manuscript entitled "Venomous Noodles: evolution of toxins in Nemertea through positive selection and gene duplication" submitted by Authors, which is a very interesting study. The authors investigated the molecular evolution of toxins in Nemertea and confirmed the positive selection and gene duplication in all investigated toxin genes. The writing framework of the paper is very good. In the introduction part, the author describes in detail the evolutionary relationship of phylum Nemertea, the research status of toxins, and scientific hypotheses. However, the material methods section needs to specify the sample size information used.

Author Response

Thank you very much for your comments. We have added the number of individuals used at each methodology in Table 3.

Reviewer 2 Report

Current MS is devoted to the molecular evolution of toxin genes in Nemertea. Methods and Results and Discussion sections presented in the manuscript are adequate and appropriate. I do not have any major objections to the MS. I believe, that current MS is suitable for publication in Toxins journal. See below for my major and minor propositions/concerns:

Minor

Line 57. Mucus of ribbon worm also contains TTXs. See:

Ali, A.E.; Arakawa, O.; Noguchi, T.; Miyazawa, K.; Shida, Y.; Hashimoto, K. Tetrodotoxin and related substances in a ribbon worm Cephalothrix linearis (Nemertean). Toxicon 1990, 28, 1083–1093.

Vlasenko, A.E.; Magarlamov, T.Y. Tetrodotoxin and its analogues in Cephalothrix cf. simula (Nemertea: Palaeonemertea) from the sea of Japan (Peter the Great Gulf): Intrabody distribution and secretions. Toxins 2020, 12, 745.

Anna E. Vlasenko, Vasiliy G. Kuznetsov, Grigorii V. Malykin, Alexandra O. Pereverzeva, Peter V. Velansky, Konstantin V. Yakovlev and Timur Yu. Magarlamov. Tetrodotoxins Secretion and Voltage-Gated Sodium Channel Adaptation in the Ribbon Worm Kulikovia alborostrata (Takakura, 1898) (Nemertea). Toxins 2021, 13, 606

Line 171-173. This phrase is not quite right. Kem (1979) extracted putative peptide toxins and characterize its property. Authors of current MS found only similarity of transcripts with domains similar to neurotoxins. Please rephrase this sentence.

Line 172. Please set reference for “Lineus longissimus

Line 255: “proboscis tegument” better use “glandular epithelium of proboscis”

Line 319-320: “Cerebratulus marginatus” is also Lineidae. Please, rephrase this sentence.

Figures 4, 5, 6, 7. Species names are not visible

Line 516 “predators A” – please set dot

Line 1030: for “PSEUDOALLELISM AND GENE EVOLUTION” change to lowercase.

Author Response

Current MS is devoted to the molecular evolution of toxin genes in Nemertea. Methods and Results and Discussion sections presented in the manuscript are adequate and appropriate. I do not have any major objections to the MS. I believe, that current MS is suitable for publication in Toxins journal.

Thanks for your positive response. Please find the detailed responses below and the corresponding corrections in track changes in the re-submitted file.

Line 57. Mucus of ribbon worm also contains TTXs. See:

Ali, A.E.; Arakawa, O.; Noguchi, T.; Miyazawa, K.; Shida, Y.; Hashimoto, K. Tetrodotoxin and related substances in a ribbon worm Cephalothrix linearis (Nemertean). Toxicon 1990, 28, 1083–1093.

Vlasenko, A.E.; Magarlamov, T.Y. Tetrodotoxin and its analogues in Cephalothrix cf. simula (Nemertea: Palaeonemertea) from the sea of Japan (Peter the Great Gulf): Intrabody distribution and secretions. Toxins 2020, 12, 745.

Anna E. Vlasenko, Vasiliy G. Kuznetsov, Grigorii V. Malykin, Alexandra O. Pereverzeva, Peter V. Velansky, Konstantin V. Yakovlev and Timur Yu. Magarlamov. Tetrodotoxins Secretion and Voltage-Gated Sodium Channel Adaptation in the Ribbon Worm Kulikovia alborostrata (Takakura, 1898) (Nemertea). Toxins 2021, 13, 606

Thanks for the suggestion. These were included in the non-toxins. Nevertheless, we included more details about the TTX in the introduction (lines 66-77).

Line 171-173. This phrase is not quite right. Kem (1979) extracted putative peptide toxins and characterize its property. Authors of current MS found only similarity of transcripts with domains similar to neurotoxins. Please rephrase this sentence.

You are correct, we did not characterize the found proteins, differently from Kem and Jacobson. We have put that in evidence in our manuscript. Nevertheless, we still discussed that these transcripts could potentially be neurotoxic (lines 200-203).

Line 172. Please set reference for “Lineus longissimus”

Thanks for pointing that out. It was set.

Line 255: “proboscis tegument” better use “glandular epithelium of proboscis”

We have made the substitution.

Line 319-320: “Cerebratulus marginatus” is also Lineidae. Please, rephrase this sentence.

We have rephrased the sentence, changing Lineidae to Lineus (and Riseriellus) genus.

Figures 4, 5, 6, 7. Species names are not visible

Thanks for pointing that out. This issue resulted from loss of resolution during the submission, which doesn’t allow Figures with higher resolution in the doc file. This should be addressed when making the final PDF with the full size pictures. In the meantime, please find attached the full sized pictures, with the visible names.

Line 516 “predators A” – please set dot

Dot set.

Line 1030: for “PSEUDOALLELISM AND GENE EVOLUTION” change to lowercase.

Change made.

Reviewer 3 Report

The ms combines transcriptomic and proteomic analyses of three nemertean species and constructs a toxin phylogeny for several nemertean toxins and putative toxins. Toxin gene duplications and positive selection processes are found.

While the ms applies this t-p approach well, from the beginning it ignors the presence of non-protein alkaloid toxins in most hoplonemerteans that have been examined as well as the lack of evidence for peptide toxins in this group. T-p methods could have been extended to look for enzymes involved in alkaloid biosynthesis in hoplonemerteans. In this regard, a recent Toxins paper demonstrating the presence of the alkaloid anabaseine in a polystyliferan hoplonemertean should be cited, since it summarizes what has been found by biochemical methods in various nemertean species and provides phylogenetic insights on toxin evolution.

Author Response

Thanks for the comments on our manuscript. We are aware of the importance of non-protein toxins and now we mentioned them in lines 63-66 and at the discussion. In this manuscript, we choose to focus on proteinaceous toxins as these are easier to detect using transcriptomic approaches, can have their evolutionary rate accessed using dN/dS measures and have been widely studied by other groups using transcriptomic approaches. We believe that an investigation of the non-protein toxins would be an interesting topic for future research. We agree, however, that the importance of these toxins to Nemertea may have passed unnoticed in the introduction, therefore, we have included a paragraph to present these toxins and their importance in Nemertea (lines 67-80).

Reviewer 4 Report

I am not an expert on toxins or the transcriptome, and can only critically evaluate the part that concerns the systematics and phylogeny of nemerteans. My comments are small and are given below.

Lines 28-29. The synapomorphy of the group is the presence of an eversible proboscis housed in a cavity called the rhynchocoel.

This is a controversial statement. We still do not know how to interpret the absence of a proboscis and rhynchocoel in Arhynchonemertes axi. Riser assumed that this nemertean initially lacked a trunk and rhynchocoel.

 Lines 31-33.  Recently, it was proposed that the phylum should be divided into three classes:  Hoplonemertea, Palaeonemertea, and Pilidiophora, the latter comprising the order Heteronemertea, and the family Hubrechtidae [3].

 First, there is a newer work (Chernyshev, 2021) that reflects the system of nemertean orders adopted in WoRMS. Secondly, the correct name is Hubrechtiidae (not Hubrechtidae), but authors should use the valid name Hubrechtellidae, which was proposed 20(!) years ago and accepted in WoRMS. Thirdly, in the WoRMS system, Pilidiophora is divided into two orders - Heteronemertea and Hubrechtiiformes. I understand when the last name is not mentioned 20 years ago, but now there is no doubt that these are two sister taxa of equal rank. Finally, it is not clear why orders are mentioned only for the class Pilidiophora? This also applies to Figure 1.

Lines 51-52. Palaeonemertea and Pilidiophora lack any rigid piercing structure.

This is not entirely true. Species of the genera Callinera and Hoploenopleus have proboscis armature.

 Lines 54-55. toxins can be found in the mucus surrounding the animal's body, acting as a potential defense mechanism against predators. Such mucus was demonstrated to contain peptides with neurotoxic and cytotoxic properties [12,13,21].

Authors should cite recent article “Tetrodotoxins Secretion and Voltage-Gated Sodium Channel Adaptation in the Ribbon Worm Kulikovia alborostrata (Takakura1898)”

 Lines 99-100. Nemertopsis pamelaroeae (Mendes et al. 2021).

For this species, the authors should not be enclosed in parentheses.

 Line 114-115. duplications in Pilidiophora than in other clades

The authors should justify that duplications occur specifically in Pilidiophora, and not in Heteronemertea and Lineidae. These articles look like duplications are a feature of Lineidae.

Line 140-141. The relationship of the major clades (Palaeonemertea, Pilidiophora and Hoplonemer-140 tea) remained as found in previous phylogenetic inferences based on genetic data (Figure 1) [4,54].

Here it is not entirely appropriate to refer to the article of Andrade et al., 2012, where the monophyly of Pilidiophora was not confirmed.

Line 129. Phylogenomic analysis

The authors discuss the position of Riseriellus occultus, but do not address the more important question: why does Lineus sanguineus not form a clade on the tree? Based on the tree, there are genetic differences between the specimens from Argentina, Brazil and France. How significant are these differences?

 References

In many articles, the authors did not italicize the Latin names of species and genera.

Author Response

Please see the reponse on the attached pdf.

Reviewer 5 Report

The manuscript titled "Venomous Noodles: evolution of toxins in Nemertea through positive selection and gene duplication" in describes thorough analysis of transcriptomes Lineus sanguineus, Nemertopsis pamelaroeae, Ototyphlonemertes erneba and N. berthalutzae ribbon worms. Authors deploy a wide variety of bioinformatic methods as well as LC/MS-based proteomics to gain understanding of putative protein toxins evolution within Nemertea. Analysis also includes previously published worm transcriptomes which places current manuscript in the context of fundamental evolutionary research.

The study seems to be rigorously planned and conducted, supplementary materials contain a lot of info useful to understanding of the paper. For me the main weak point of the manuscript is a little bit tough style of presentation. The paper would benefit from polishing graphics and proper graphical abstract.

Author Response

"The manuscript titled "Venomous Noodles: evolution of toxins in Nemertea through positive selection and gene duplication" in describes thorough analysis of transcriptomes Lineus sanguineus, Nemertopsis pamelaroeae, Ototyphlonemertes erneba and N. berthalutzae ribbon worms. Authors deploy a wide variety of bioinformatic methods as well as LC/MS-based proteomics to gain understanding of putative protein toxins evolution within Nemertea. Analysis also includes previously published worm transcriptomes which places current manuscript in the context of fundamental evolutionary research."

Thanks for your comments, we appreciate that.

"The study seems to be rigorously planned and conducted, supplementary materials contain a lot of info useful to understanding of the paper. For me the main weak point of the manuscript is a little bit tough style of presentation. The paper would benefit from polishing graphics and proper graphical abstract."

Thanks for pointing that out. To address this issue, we have made a Graphical abstract including the methodology and main results of our manuscript. We also changed the Figs 2-7 to make them clearer.